# The Core Value of Sustainable Fashion: A Case Study on "Market Gredit"

Young Kim [1] and Sungeun Suh [2],*

1  Gredit Inc., 405, Seoul Upcycling Plaza, 49 Jadongchasijang-gil, Seongdong-gu, Seoul 04807, Korea
2  Fashion Design Major, College of Arts, Gachon University, Seongnam-si 13120, Korea
*  Correspondence: sesuh@gachon.ac.kr; Tel.: +82-31-750-5948

**Abstract:** Sustainability and ethical consumption have emerged as essential topics globally in the carbon-neutral era. The adoption of eco-friendly production and distribution methods have been prominent in the fashion industry as well. This study performed a qualitative case study analysis based on a literature review and in-depth expert interviews. The search yielded ten keywords reflecting how value producers pursue sustainable fashion. These keywords are "eco-friendly", "recycled", "vegan", "upcycled", "zero waste", "local production", "fair trade", "women-owned", "gives back" and "animal protection". The values are divided into five core environmental values and five core social values. Based on in-depth interviews with domestic sustainable fashion experts, the core values and detailed standards of sustainable fashion were specified and presented as practical values. These values were also applied to the Market Gredit platform, an integrated e-commerce platform for sustainable fashion brands in South Korea. This research contributes to improving the environmental and social impacts throughout the value chain of the fashion industry by presenting the core values and detailed standards of sustainable fashion suitable for communication with consumers.

**Keywords:** sustainable fashion; core environmental values; social core values; lifestyle platform; Market Gredit; value consumer; value consumption

## 1. Introduction

In this carbon-neutral era, sustainability and ethical consumption have emerged as salient topics globally. On the industry front, it has become essential to adopt the environmental, social, and governance (ESG) management paradigm. This adoption has been prominent in the fashion industry. According to the firm Research and Market, the global conscious fashion market is expected to grow from USD 5.84 billion in 2021 to USD 8.3 billion by 2025 [1]. However, according to a Harvard Business Review report [2], despite the recent increase in fashion companies' adoption of environmentally friendly production and distribution methods, they still lag in their practices. This can be attributed to the structural characteristics of the fashion industry, namely, the complex supply chains and the inevitability of large amounts of inventory. According to Professor Pucker's research team, less than 1% of recycled clothing is currently converted into new clothing worldwide. Furthermore, the rate of carbon emissions reduction from buying and selling used goods has averaged less than 0.01% per year over the past decade [3]. The head of the Massachusetts Institute of Technology's Center of Transport & Logistics (MIT CTL), Yossi Sheffi, investigated the environmental impact of the famous sports brand Nike on the global supply chain [4]. The study found that Nike produced 16,000 materials through 1500 global partners, and 56% and 83% of the carbon dioxide produced and water used, respectively, by Nike, came from its raw material value chain. Apart from the environmental issues, the fashion industry is also rife with other social problems. For example, the Rana Plaza incident in Bangladesh (2013) highlighted social problems such as child labor, forced labor, discrimination, occupational safety and health, and diversity within the fashion value chain [5].

Several studies have pointed out the serious environmental and social hazards caused by the fashion industry. However, the industry is transitioning toward a new sustainable and ethical fashion paradigm. Sustainable consumption is a recent trend where consumers consider the social and environmental aspects [6]. It must be noted that the ongoing pandemic has led to an increase in consumers' pursuit of eco-friendly consumption. According to Accenture [7], a global consulting firm, consumers are shopping mindfully and cost-consciously, with demand for local, sustainable and value brands rising. Accenture [7] also said that 61% of consumers have started making eco-friendly purchases since the pandemic. Even after the pandemic, 89% of consumers said they would maintain this consumption pattern. According to McKinsey [8], 90% of generation Z, a class of next-generation consumers, believe that brands must articulate their stance on environmental issues in the fashion industry. It is important to practice and communicate sustainable practices meeting consumer needs.

In the context of practices, it is difficult to have an insight into the entire supply chain. Owing to the complex structure of the value chain in the fashion industry, experts have provided different interpretations of sustainable fashion. Although public interest and awareness of the environment have increased, the reality is that some consumers still do not understand the term "sustainability". Some feel skeptical about the greenwashing of companies, some perceive eco-friendly fashion to be expensive, and some share a reluctance to use recycled products [9]. These contradictions keep consumers from changing their existing consumption practices.

The purpose of this study is to derive sustainable keywords that could have an impact on the entire fashion supply chain through an analysis of the status of producers and an in-depth literature research, and ultimately to communicate at the consumer's eye level. This study improves environmental and social impacts by focusing on Market Gredit [10]. It is an integrated e-commerce platform for domestic sustainable fashion brands and focuses on the case of branded stores in Korea. The study also proposes sustainable core values for fashion, which are applied using the Market Gredit platform. The platform discovers and distributes fashion brands that fit each core value type. Thus, the study helps consumers to understand sustainable fashion and, in the process, contributes toward revitalizing the market.

## 2. Theoretical Background

### 2.1. Status of Sustainable Fashion from a Value Chain Perspective

Jeon, Han, and Go [11] introduce the fashion industry value chain as an expanded concept, including consumer use, recycling, and disposal after production and distribution. This value chain comprises the manufacturing and sourcing of raw materials and yarns; weaving, processing, and sourcing of fabrics; planning and design; production, sale, and distribution; and use, disposal, recycling, or upcycling. Kim [12] presents ethical fashion from the perspective of the value chain of fashion design and demonstrates its role in reducing the environmental impact in the entire process—from production to distribution. Choi and Lee [13] define ethical fashion as the production of goods by humans using natural products and energy without harming the environment. Jestratijevic and Rudd [14] identify six types of sustainable fashion—biodegradable, recycled products, new luxury products, used and vintage products, repaired/upcycled/upgraded products, ethical products, and products with public accreditation.

According to Jeon, Han, and Go [11], sustainable fashion research has focused on the individual stages of the product life cycle, such as materials, design, production, purchasing, disposal, and upcycling. However, it has been limited in terms of product life management. Karell and Niinimäki [15] conducted in-depth interviews with five sustainable fashion experts and 31 designers. They dispel the notion that most of the participating designers' approaches to sustainability are limited to fragmented methods such as using eco-friendly materials or upcycling. They show that fashion companies prioritize sustainability in the design process, especially during material selection, and in the subsequent processes

focusing on product durability, timeless designs, fit for purpose, zero waste, upcycling, repairability, and recyclability. However, most experts emphasize a sustainable plan from the production inception phase to the recycling phase after usage from the circular economy perspective. This requires empathy and cooperation from not only producers but also consumers. Sustainable fashion has evolved into a concept including both ethical and societal aspects. It embraces culture-, time-, and value-related perspectives by focusing on the issues of labor rights, fair trade, the life patterns of consumers, and their values of happiness [6] (p. 13).

On the industry front, international organizations have been proactively discussing sustainability and environmental regulations. This is attributed to the industry's need to embrace the adoption of sustainability and conform to sustainability benchmarks. The sustainability index has become an important indicator of the management method of a fashion company. The index plays an important role in the systematic and structural changes in the fashion industry [6] (p. 18). A representative sustainability evaluation index is the Higg Index, established in 2012 by the Sustainable Apparel Coalition (SAC) in the United States. The SAC aims to help businesses make an exponential impact. The Higg Index is a suite of tools that facilitate the standardized measurement of value chain sustainability. It consists of five instruments: the Higg Facility Environmental Module (Higg FEM), the Higg Facility Social and Labor Module (Higg FSLM), the Higg Brand and Retail Module (Higg BRM), the Higg Materials Sustainability Index (Higg MSI), and the Higg Product Module (Higg PM). They assess the social and environmental performance of the value chain and the environmental impact of products [16,17]. In addition to the Higg Index, the Fashion Transparency Index was created in 2017 by the UK's Fashion Revolution Organization and the Sustainable Brand Index was created in 2011 in Sweden, which measure brand awareness of sustainability [6]. Patagonia is a good example of a fashion brand that endeavors to improve its environmental and social impacts by controlling the use and disposal of raw materials. Patagonia is a member of the SAC and a leader in developing the Higg Index. Seo and Yoo [17] also pointed out how Patagonia's participation in the Higg Index has contributed to improving the environmental impact of the entire fashion supply chain. The study also showed Patagonia's commitment to corporate social responsibility (CSR) through its '1% for the planet' foundation, which donates 1% of the corporate sales to environmental groups.

The domestic fashion industry also focuses on ESG management centered on eco-friendly values. The recycled polyester fiber market is steadily growing, and the market is witnessing the increased entry of fashion companies using vegan materials. Fashion companies are adopting various methods to enhance the sustainability of their products, thereby reducing carbon emissions. These methods include using eco-friendly yarns, producing eco-friendly packaging and clothing tags, and recycling the last season's inventory [18]. With consumers' value consumption becoming mainstream, a company's contribution to the environment and society has become a major purchasing factor. Concerning the sustainable methods adopted by Korean companies, recently, the Korea Corporate Governance Service evaluated the ESG rating as a company's sustainability indicator [19]. The following section discusses domestic fashion companies that received the top ESG ratings. 'Records' of the KOLON Fashion in Culture is an upcycling brand that proposes solutions using inventory held over ten years. 'Records' employs North Korean defectors, single mothers, refugees, and people with disabilities to dismantle and reassemble clothing and carry out activities consistent with the ESG paradigm [18].

Shinsegae International's lifestyle brand *JAJU* became the first Asian company to secure an exclusive license for "Cotton Made in Africa" (CmiA). It is introducing high-quality eco-friendly products [18]. LF introduced the 'Upcycling Project Line' through the brand Daks, which cuts and pastes shirts and pajamas in the inventory and re-commercializes them. Hazzys makes sneakers made from apple peel. Young casual brand 'Atcorner' developed a non-toxic dye for its eco-friendly denim line [19]. On 21 July 2020, the fashion industry saw its first eco-friendly packaging system 'CartonWrap' [18]. Samsung C & T

Fashion Group 'Beanpole' introduced a knitted bag made from recycled plastic bottles; it also released a wooden bag that obtained an eco-friendly quality certification from an American regulatory and safety standards organization (Green Guard) [18]. Athleisure wear 'Mulawear' has been discovering and using eco-friendly yarns and biodegradable and breathable materials, such as modal using beech extract and Tencel using eucalyptus wood raw materials [18].

From a value chain perspective, the status of sustainable fashion in Korea and abroad is that the largest companies are developing eco-friendly products or lines for their existing brands and implementing eco-friendly values based on feasibility. However, they face challenges in pursuing sustainability from an integrated perspective and penetrating the entire supply chain. Small eco-friendly brands struggle to establish a system to obtain certifications and ratings, including ESG. Owing to inadequate distribution channels, it is difficult for these brands to promote and sell their brand and products to consumers. Social venture companies that manufacture and distribute sustainable products have a high-cost structure, ranging from raw material sourcing to production and manufacturing. They also face constraints and uncertainties in distributing the products. In addition, the cost of these products is higher than that of other generic competing brands. This aspect hinders the expansion of the base of their current production and distribution structure. Therefore, it is necessary to create platforms and physical spaces that can bridge the gap between sustainable fashion and "good" consumption [20].

### 2.2. Value Consumer Perception of Sustainable Fashion

Lundblad and Davies [21] analyze UK consumers' motives for purchasing sustainable fashion and identify six patterns in their motives. They buy less, comfort and good looks, health, environmental consciousness, a sense of accomplishment through eco-friendly fashion consumption, and ethical consumption (purchasing brands produced without inducing inequality). To popularize sustainable fashion among other consumer groups, it would be necessary to expand value chain sustainability from the usage of the product to its disposal. It will also be necessary for sustainable fashion to consider environmental and social factors simultaneously to become more competitive in the market.

Wiederhold and Martinez [9] conducted an in-depth consumer interview on why German consumers' interest in environmental issues is unrelated to value consumption. They found several reasons for such behavior. One is the perception that eco-friendly fashion is outdated and expensive, there is a lack of information to identify ethical fashion brands and distinguish them from greenwashed products, ethical fashion brands are inaccessible, it is difficult to change consumption habits, and that change in one person's behavior may not create a significant impact. To further the growth of the sustainable and ethical fashion market, they [9] suggested using social media platforms to inform customers about the suitability of eco-friendly products in terms of affordable pricing, attractive styles, and an accessible distribution network. The study by Lee and Kim [20] also pointed out the unstable distribution and low consumer awareness of sustainable fashion products. While consumer response to sustainable fashion products reflects their social-value orientedness, they tend to prefer trendy clothes. Furthermore, there is a perception that recycled clothes should not be expensive. Some consumers also perceive that the elements of good consumption become auxiliary when waste is recycled to create a product. In other words, they believe that good consumption is not actively practiced when recreating a product from waste.

In this context, it is useful to focus on supply chain management (SCM) studies that Cottrill [22] (the editorial director of MIT CTL) introduced to MIT. These studies focused on improving the environmental impact of processes in each stage of the consumer goods supply chain: production, purchase, use, and disposal. Among them was a study on the environmental impact of e-commerce companies and distribution in Mexico. In this study, after the surveyed customers learned about the environmental impact of fast delivery, 70% of the surveyed customers expressed their intention to extend the expected delivery date by

an average of 4 days. Specifically, instead of informing the customers about the rise in the carbon footprint of fast delivery, the study explained that an eco-friendly delivery option could be equivalent to planting trees. This led the consumers to opt for the eco-friendly shipping option. This proves that when communication on the environmental impact of sustainable consumption aligns with the consumer perspective, it elicits a positive consumer response. According to Sheffi [4], Patagonia also recognized that prior to the development of the Higg Index, that is, when the company still had a large carbon footprint, transparency regarding the environmental impact of each stage of the value chain led to consumer confusion. They developed the Higg Index to make communication understandable to consumers. However, the various environmental indexes, including the Higg Index, have not yet reached the stage of popularization. Of the experts and designers interviewed by Karell and Niinimaeki [15], three have worked with materials rated by the Higg Index, and four have only attempted to use such materials. Concerning the Index, 15 had heard of such indices, and the remaining nine were unaware of the indices. Global fashion brands have been applying the sustainability index, buying raw materials, and creating various certification labels according to the sustainability policies implemented by each country. However, this study highlighted the need for easy selection criteria that consumers can trust and purchase. It also highlighted the need for all the companies in the supply chain to collaborate and commercialize an advanced index such as the Higg Index, given that it entails high costs and resources. Thus, the study believes that such indices may not be launched in Korea within a short period. Accordingly, the use of eco-friendly materials, upcycling, zero-waste, representation of the underprivileged communities in the employee base, and fair trade are necessary to establish an index that can be easily accessed by the domestic value producers and value consumers of sustainable fashion. Furthermore, it is necessary to start by categorizing the environmental and social improvement factors at each stage of the value chain. This classification is very important in that it can work as a basis for communication that meets consumers' understanding by presenting specific standards and guidelines throughout eco-friendly fashion.

Concerning consumers' purchase decisions, according to the 2020 McKinsey report [23], of surveyed consumers, 67% consider the use of sustainable materials to be an important purchasing factor, and 63% consider a brand's promotion of sustainability in the same way. According to a study by the LIM College [24], a fashion business college in New York, about 90% of the surveyed millennials believed that their value consumption enables businesses and governments to break free from traditional practices and become more sustainable. Among these millennials, only 34% of the respondents said that they had purchased ethical fashion products. The key factors influencing their purchasing decision were ease of purchase (95%), price/value (95%), product uniqueness (92%), brand recognition (60%), and sustainability (35%). Rachel Arthur, the Chief Information Officer of The Current, a fashion innovation consulting firm, noted that although consumers are environmentally conscious, most want to buy a good product rather than have a detailed understanding of the supply chain [25]. Experts say a brand's simple and powerful messaging and quality induce consumer purchases. Therefore, it is necessary to write detailed descriptions of a product's environmental and ethical certification in online transactions or to attach an ethical fashion certification label to the product so that consumers can easily access information about sustainable fashion products. In addition, it is important to provide consumers on e-commerce platforms with convenient access to sustainable fashion by providing a filter function for ethical products, similar to filters for price, brand, or color. These factors are important, given that ethical fashion is moving from the realm of marketing and corporate communication to the realm of consumer value [26].

This study demonstrates that, to promote value consumption for sustainable fashion, the fashion supply chain needs to be upgraded in terms of the whole system and not just through partial improvements at the environmental and social viewpoints of sustainable products. This is necessary so that companies can directly communicate with consumers about the complex environmental and social impacts.

### 2.3. Cases of Value Consumption Platforms That Connect Value Producers and Consumers

The rising interest in environmental and social issues has increased the number of value producers and consumers globally. This factor has led to a steady rise in fashion and consumer goods platforms connecting the producers and the consumers. A good example of an overseas online platform is the US site Donegood [27]. Donegood is designed to introduce brands and products based on ten different environmental and social core values and to search for items of interest by core values, categories, and brands. Donegood operates more than 500 brands based on 17 fashion and household goods categories. It is positioned as an aggregate platform of value consumption and is nicknamed the "Amazon of good companies" in the United States [28]. Donegood's ten core values are eco-friendliness, worker empowerment, vegan, women/person of color-owned, toxin-free, giving back, recycled/upcycled, organic/genetically modified organisms-free, cruelty-free, and made in the United States [27].

Concerning domestic online and offline shopping malls, a representative example of an offline platform is the Sustainable Ethical Fashion Hub. It is a fashion project supported by the city government of Seoul. Another example is the Seoul Ethical Fashion (SEF) store run at the Condemn DDP. SEF stores select fashion companies and brands practicing environmental, social, and ethical values. This approach toward sustainability enhances their skill set and promotes qualitative growth. In SEF stores, the brands sell their products through multi-shops; currently, roughly 40 social venture brands are part of the store [29]. Another example is 'The Each' store operated by Art Impact, a social enterprise that is part of the JDC Duty-Free, at the Jeju domestic duty-free stores. These stores are smaller than the SEF stores; they currently sell 15–20 brands [30]. Each store also runs an online store. It serves as a platform for sustainable production and consumption by focusing on the three core values of sustainable production and consumption, climate change response, and marine ecosystem conservation—all of which are part of the United Nation's sustainable development goals (SDGs) [31]. Each store only allows products from companies certified as social enterprises. There has been a rise in the number of offline value consumption-based stores. They have appeared as zero-waste shops centered on household goods rather than fashion. A representative example is Almang Market in Mangwon-dong, Seoul, which was established with the aim of reducing the use of plastic containers. It is the first refill store in Korea and allows consumers to make unlimited purchases in bulk containers [32].

In the case of domestic online stores, most of the stores are small, zero-waste, household goods stores. Examples of platforms also selling fashion items alongside household goods include the 'Emulgun Market' [33] and 'More Store' [34]. The 'WeDo stores' of the Kolon Mall [35], 'Green Friends Hall' of Hyundai [36], and other large retailers are also riding the eco-friendly trend. There has been a rise in ethical fashion corners in shopping malls. Nevertheless, it is very difficult to establish a distribution channel. It must be understood that offline distribution is limited, and that online fashion distribution requires lookbook shooting and marketing [20] (p. 95). In addition, eco-friendly fashion-related platforms tend to assort items based on brand recognition or fragmentary standards instead of categorizing products in terms of environmental and social impacts from an integrated point of view of the value chain. Given the rise in the ESG management and value consumption trends in the fashion industry, this study focuses on the need for information that helps consumers verify sustainable fashion products, communication standards allowing the value pursued by the product to be easily communicated to consumers, and the need for a professional distribution platform.

## 3. Research Methods and Procedures

### 3.1. Research Methods

The researchers conducted a qualitative case study based on a literature review and in-depth expert interviews as the research model (Figure 1). A qualitative case study is a method of in-depth investigation of single or multiple limited cases through various qualitative methods. The researcher observed, interviewed, and participated in various

formal and informal forms of document collection. Data collection led to an in-depth analysis of the cases studied and the development of concepts and theories based on the cases [37,38]. In accordance with Yin [39], this study conducted an explanatory case study. An explanatory case study is conducted to provide a more in-depth and sufficient explanation based on inductive analysis or theoretical sampling by examining several similar cases [38].

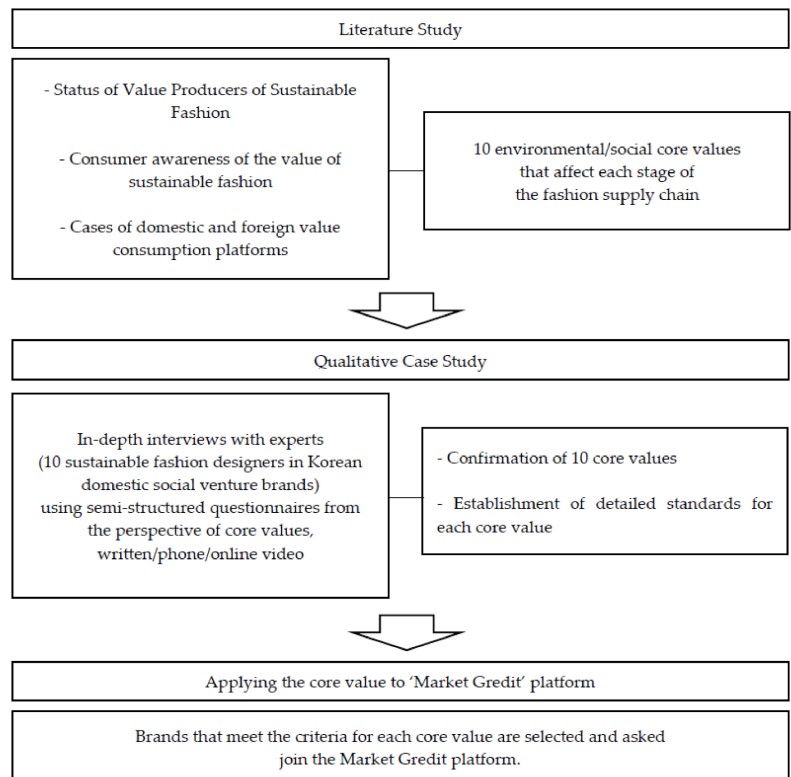

**Figure 1.** Research model.

For the case study, it was important to develop a wide range of data collection procedures based on various information sources. Although new types of qualitative data are continuously emerging in qualitative research, they can be categorized under four basic groups. These groups are as follows: interview (one-on-one interview, face-to-face interaction in a group, web-based interaction), observation (non-participatory and participatory observations), documents (private and public documents), and audiovisual materials (artifacts in photographs) [37,40]. In this study, data were collected and analyzed through semi-structured questionnaire interviews. The expert interviews were conducted as one-on-one online interviews as qualitative research, and a non-real-time method was used in which the researcher sent questions to the participants by e-mail, and the participants responded. This is an efficient method that empowers the participants as it gives them sufficient time to think about their responses. In addition, it enables interviews with participants who cannot meet in person due to physical distance and time limitations [38,41]. Therefore, this method was appropriate because most of the questions posed to the interviewees in the questionnaire of this study require sufficient thought and time rather than improvised answers. In cases of insufficient information provided in the first e-mail response, additional inquiries were made through a second e-mail or video conference.

Before conducting the survey, the study analyzed prior research—articles and cases of domestic and foreign value consumption platforms—to understand the status of sustainable fashion practice. The search yielded ten keywords reflecting how value producers pursue sustainable fashion. The keywords are "eco-friendly", "recycled", "vegan", "upcycled", "zero waste", "local production", "fair trade", "women-owned", "gives back"

and "animal protection". The study also analyzed how these core values impacted each vertical value chain stage, including textile material production, dyeing and processing, planning and design, sourcing and production transits, distribution and sales, and usage and disposal. These values were classified into five core environmental and social core values. In the evaluation process, the researcher reviewed the impact of the first ten keywords on each stage of the fashion supply chain, and the detailed items of related goals that are part of 17 UN SDGs [42]. These were poverty eradication, zero hunger, health and well-being, quality education, women's equality, clean water and sanitation, reasonable clean energy, industry, innovation and infrastructure, inequality reduction, sustainable cities and communities, responsible consumption and production, climate action, life under water, hotbeds of life, peace, justice and strong institutions, partnerships with objectives. Table 1 shows the sub-items of environmental and social values, and the ten core sustainable values are indicated in the parts corresponding to the value chain of the fashion industry.

**Table 1.** Core Values of Sustainable Fashion.

| Vertical Value Chain Core Values | | Material Production | Dyeing and Processing | Planning and Design | Transit | | Distribution and Sales | Usage and Disposal | Detailed Criteria |
|---|---|---|---|---|---|---|---|---|---|
| | | | | | Sourcing | Production | | | |
| Environmental values | Eco-friendly | ○ | ○ | | | | | ○ | - Natural fibers<br>  • organic cotton, hemp, and ethically produced wool<br>  • eco-friendly regenerated fibers such as Tencel<br>- (vegetable) Natural dye |
| | Recycled | ○ | ○ | | | | | ○ | - Recycled polyester or nylon<br>- Dyeing techniques that reduce water use, such as E-dye<br>- Recyclable after disposal |
| | Vegan | ○ | ○ | | | | | | - Owner leadership to practice vegan life<br>- Use of non-animal eco-friendly fibers<br>- (For leather) vegetable processing<br>- (For cosmetics) Exclusion of animal testing |
| | Upcycled | ○ | ○ | ○ | | | | | - Upcycling design/production using waste materials it is different from recycling in that it is designed/manufactured by preserving the original materials of products |
| | Zero waste | | | ○ | | ○ | ○ | ○ | - Product: pattern method with less than 5% waste<br>- Packaging: plastic-free packaging |
| Social values | Local Production | ○ | ○ | ○ | | ○ | | | - Local production through the (socially vulnerable class or social economy organization) |
| | Fairtrade | ○ | ○ | ○ | | ○ | | | - Third World Fair Trade Products |
| | Women-owned | | | ○ | | | | | - Female ownership, Companies with 50% or more women<br>- Companies employing women from disadvantaged groups |
| | Givesback | Profit redistribution resulting from the entire value chain from raw materials to distribution | | | | | | | - Regular sponsorship to organizations that attempt to solve environmental and social problems |
| | Animal protection | Sharing/donation activities focused on animal protection | | | | | | | - Supports activities such as regular sponsorship of 'animal protection groups and a leadership that practice vegan life' |

Based on the ten core values of sustainable fashion (Table 1) derived from previous studies, this study first focused on 40 domestic social ventures based on improving environmental and social issues. It did not focus on large corporations that have recently started to embrace eco-friendly fashion as a trend. The study interviewed ten designers as brand owners who are leading sustainable fashion. All researchers and interviewees were South Korean, so the interview language used was Korean. In the case of the ten keywords, as illustrated in Figure 2, English/Korean were communicated in parallel. In addition, most of the ten English keywords are terms used routinely by workers in the South Korean fashion industry.

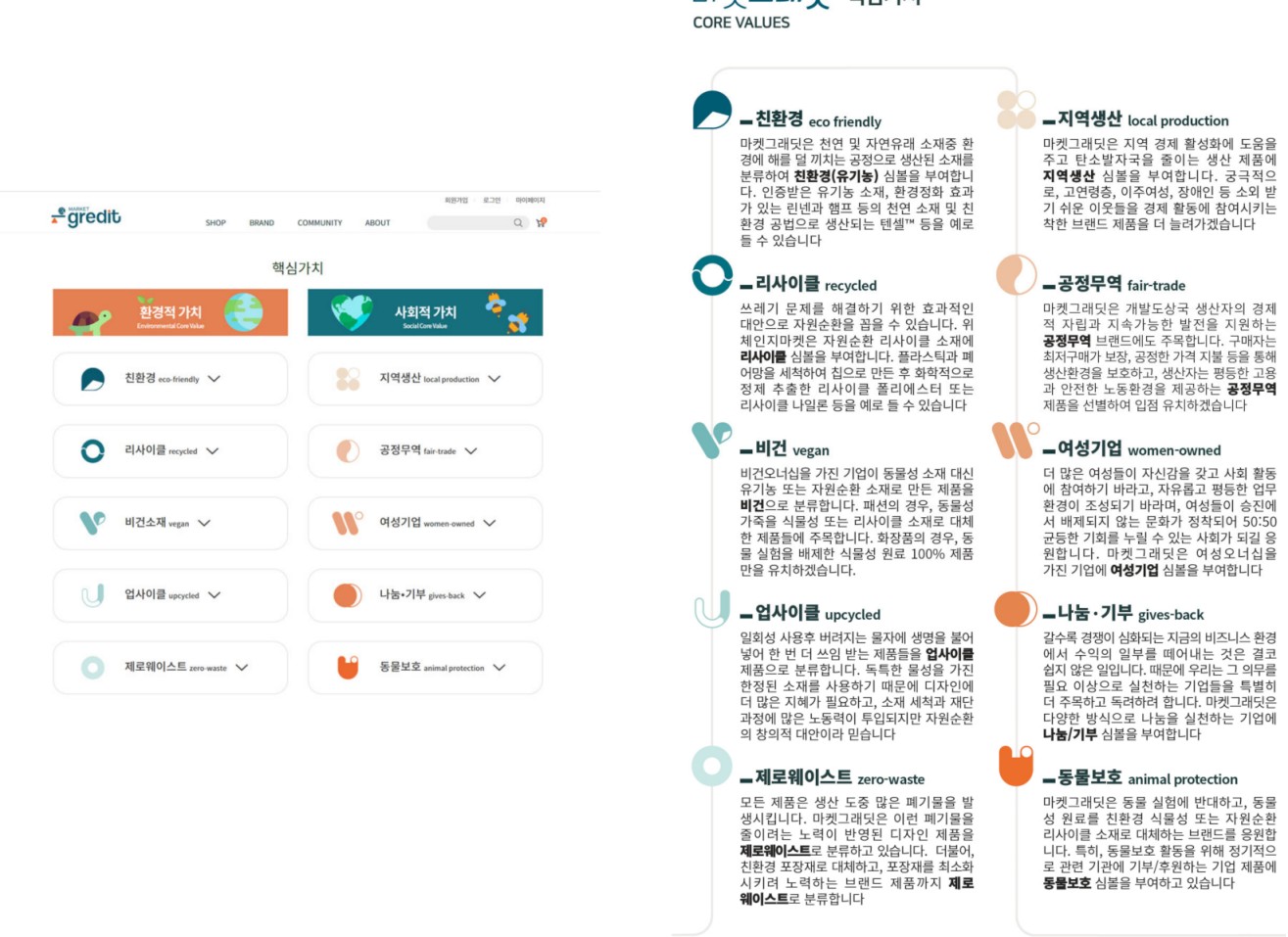

**Figure 2.** Sustainable core values posted on the website of Market Gredit. Source: https:ans//gredit. io/custom/common/core_value.html?cate_no=229 (accessed on 10 July 2022).

This study identified the core values and practices each brand pursues for sustainable fashion. Thus, this study attempted to re-establish ten different detailed standards for environmental and social values that affect the value chain. In addition, selection criteria for brand and product selection were created using the example of Market Gredit, which has established a national platform for sustainable fashion and lifestyle based on these ten core values and detailed standards. By labeling each brand's sustainability core values and practices, this study sought to create a milestone to distinguish products that strive for comprehensive improvement from the consumer's perspective.

### 3.2. Collection of Data from Research Subjects

In qualitative research, we use the concept of intentional sampling. This means that the researcher selects the sites and people needed for the research. This is done because it can

provide customized information about research problems and key phenomena [37]. In this study, in-depth interviews were conducted by selecting ten fashion brand designers who met at least one of the five criteria. This approach was adopted to facilitate the selection of a representative domestic sustainable fashion company as the research subject. The first criterion was that the brands should belong to the sustainable ethical fashion (SEF) store operated by the Sustainable Ethical Fashion Hub (SEFH) and supported by the Seoul City government. Second, the brands should have experience in participating in well-known fashion fairs and events at home and abroad. Third, the brands should have won various competitions. Fourth, the brands should belong to companies that use sustainable design and sustainable materials technologies (patents). Fifth, the brands should be well-known at home and abroad, such as Amazon.com, Musinsa, W Concept, 29 cm, and Ideas.

In the designer selection stage, 40 brands were reviewed based on their operating period, within a broad spectrum from 2 to 14 years. Prior to the interview, ten core values of sustainable fashion derived from prior research and case studies were selected (Table 1). In this study, we attempted to select diverse brands that practice the ten sustainable values, and Table 2 illustrates the most representative core values that best represent the brand's identity among the various core values corresponding to each brand. In terms of categories, the final ten brands were determined by including clothing-oriented and miscellaneous goods-oriented brands in a balanced way. Information about the participants was obtained from the interviews (Table 2). The ten interviewees are both brand owners and designers who are the most knowledgeable about the brand's philosophy and processes and think deeply about sustainability. Their answers should, therefore, be fully representative of the company's situation. In the process of selecting interviewees, it was also confirmed that they had sufficient practical knowledge about the sustainable fashion industry in South Korea through informal communication.

### 3.3. In-Depth Expert Interviews

An in-depth, one-to-one interview was conducted with the research subjects. Data were collected from 24 November 2021 to 31 March 2022, when it was approved by the Bioethics Committee of Gachon University (1044396-202109-HR-187-01). Before starting the in-depth interview, the contents and purpose of the study were fully explained to the research subjects by phone and e-mail, and consent was obtained in writing before proceeding. In consideration of the COVID-19 pandemic, a semi-structured questionnaire was used during the in-depth interview. The research subjects were requested to send their responses in writing and were assured of a reply within a month of sending the responses. Later, a second request was made for the parts that needed supplementation. It included an additional common question to select the core value for which sustainable practice was implemented for each brand. For areas that were found to be lacking, additional interviews were conducted via video and phone calls. All interview contents were based on written responses prepared by the research subjects. The telephone and online video interviews were recorded with the consent of the research subjects, and the transcripts were prepared as they were recorded and analyzed together.

The questions (Table 3) used for in-depth interviews were basic survey questions. They aimed to understand the general status of the brand. The questionnaire also asked about the motivation for developing and perceiving sustainable fashion to confirm the designer's belief philosophy and sincerity toward improving environmental and social issues. Second, questions were asked to verify whether the designers were adopting sustainable practices that contribute to the environment and society from a value chain perspective. In addition, this study asked questions about the brands' communication with consumers. This question aimed to understand whether the direction of each brand's sustainable practices was acceptable not only from the manufacturers' perspective, but also from the consumers' perspective, and was able to generate a broader public consensus. Third, questions were asked about the ultimate direction and goal of being a sustainable fashion brand. Fourth, there were questions regarding the level of awareness, the need for,

and the limitations of sustainable fashion certification. Finally, the study asked questions about the conditions necessary for the revitalization of the sustainable fashion industry and questions regarding development strategies for sustainable fashion. In addition, this study did not reveal the ten items of sustainable core values (Table 1) derived by the researchers. This approach was adopted to exclude preconceived notions of the research subjects. After receiving answers to all the questions, this study asked for opinions on ten core values, which were intended to supplement/modify the detailed standards of the ten core values. The interviewees recognized the clichés and various contradictions of greenwashing and discussed in detail the core values of sustainable fashion that they conceptually know and the parts that are being practiced or can be practiced.

**Table 2.** List of study participants based on the information from the interviews.

| Designer | Brand Name (Time of Launch) | Representative Value | Representative Category | Major History |
|---|---|---|---|---|
| A (woman, 54 years old) | G: RU (2008) | fair trade | clothing | - Part of SEF stores<br>- Presented cases of the Korean Fair Trade at the 2016 WFTO Asia General Assembly (Bangkok)<br>- Acquired the international fair-trade certification in 2018 (WFTO) |
| B (woman, 36 years old) | ZERO DESIGN (2014) | Zero waste | Clothing, miscellaneous goods | - Part of SEF stores<br>- 2014 patent in zero waste pattern method technology<br>- Attended 2016 Seoul Fashion Week Fashion Show |
| C (woman, 35 years old) | CUECLYP (2016) | upcycling | Miscellaneous goods | - Part of the SEF Stores<br>- Participated in the 2016 DDP Seoul Upcycling Exhibition<br>- Participated in the 2019, 2020, 2021 Seoul Design Festival Fair |
| D (woman, 43 years old) | OPEN PLAN (2017) | vegan | clothing | - Participated in the 20SS, 21SS Helsinki Fashion Week<br>- Participated in the 19SS, 20SS, 20FW Who's Next Paris<br>- Participated in the 2020 WWF Korea Sustainable Fashion Project Re:Textile (in collaboration with IKEA) |
| E (woman, 39 years old) | NOT OURS (2017) | vegan | Clothing, miscellaneous goods | - Has stores as part of Musinsa, W. Concept,<br>- Published a book in 2022, <The Wardrobe that Saves the Earth> |
| F (woman, 50 years old) | Harlie K. (2018) | Upcycling | Misc. goods | - Owns businesses as part of Musinsa, W concept, Amazon (Amazon.com)<br>- 2018 Red Dot Design Award Winner |
| G (woman, 36 years old) | OVER Lab. (2019) | upcycling | Miscellaneous goods | - Part of SEF stores<br>- Runs business as part of Musinsa, 29 cm, etc.<br>- Cited to be Honorable Mention for 2020 K Fashion Auditions and won an award from<br>- Water Love Upcycling Contest |
| H (woman, 50 years old) | I WAS PLASTIC (2019) | recycling | Misc. goods | - Runs business in W concept, 29 cm, Amazon (Amazon.com) |
| I (woman, 45 years old) | ICONPLE (2019) | organic Eco-friendly | clothing | - Part of SEF stores<br>- Participated in the 2020 WWF Korea Sustainable Fashion Project Re:Textile |
| J (man, 45 years old) | Project 1907 (2020) | recycling | Misc. goods | - Runs businesses as part of Ideas<br>- Possesses platex technology, a recycled material made by collecting domestic plastic bottles |

**Table 3.** Composition of semi-structured questionnaires.

| |
| --- |
| **I. Investigation of Basic Brand Status** |
| **II. Motivation and awareness of sustainable fashion**<br>   (1)   Brand launching background<br>   (2)   The concept and definition of sustainable fashion from a designer's point of view |
| **III. Sustainable fashion brands from the point of view of the value chain and questions on brands' contribution to the environment and society**<br>   (1)   How they communicate their activity to the consumers<br>   (2)   Limitations felt while communicating with consumers in the way stated in (2) |
| **III. The direction and goal pursued as a sustainable fashion brand**<br>   (1)   (Even if it is currently not possible owing to various circumstances) Actionable goals that they want to reach in the future<br>   (2)   Conditions to be established internally and externally to achieve the goals mentioned in (2) (1)) |
| **IV. Awareness, necessity, and limitations of sustainable fashion certification indices**<br>   (1)   The level of understanding regarding the certification for sustainability in the fashion industry, indices, and labels, among others.<br>   (2)   Views on the necessity or limitations of such certification labels from the point of view of consumers |
| **V. Necessary conditions or development strategies for vitalization of the sustainable fashion industry** |
| **\*\*\* Additional questions asked after receiving the replies to the above questions**<br>   (1)   From the point of view of the value chain, we were able to evaluate what the brand is currently doing to contribute to the environment and society. Please select among the ten core values (organic/green, recycling, vegan, upcycling, zero waste, local production, fair trade, women's business, sharing/donation, animal protection) that best expresses the brand's practices (multiple choices are possible)<br>   (2)   In addition to the examples listed below, please make suggestions if there is a keyword that is more suitable for the brand from the viewpoint of the consumer. |

## 4. Research Results and Discussion

### 4.1. Status of Sustainable Fashion Practice from an Environmental and Social Point of View

#### 4.1.1. Concept and Perception of Sustainable Fashion

When asked about sustainable fashion, most designers emphasized the need for efforts across the value chain. Eight out of ten designers said the following. First, things can contribute to solving problems from an integrated perspective of the entire value chain from the beginning to the end of clothing. Second, a holistic view based on the understanding of the climate crisis and the plastic waste problem-from growing the materials, processing, production, transportation, design, manufacturing, and use to disposal and increasing choices that minimize environmental impacts throughout the life cycle of clothing. Third, the reduction or elimination of substances harmful to the environment in all the processes, considering the environmental impact in each stage of production/usage/disposal or the processes of material selection, sourcing, production, usage, and disposal. Meanwhile, sustainable fashion is now widely recognized as eco-friendly, and there was also the view that the social aspect cannot be excluded. This means sustainable and ethical fashion is an integral part of the environment and society. There is also a view that labor, environment, and animal welfare should be seen in an integrated way.

#### 4.1.2. Current Practices for Environmental and Social Contribution from a Value Chain Perspective

Without presenting the ten core sustainable values derived from the study, each designer was asked how they are currently contributing to the environment and society from a value chain perspective of the fashion industry. Most brands introduced specific examples

of practices consistent with the representative core values classified in the preliminary investigation stage. These were fair trade, zero waste pattern method, upcycling design, use of vegan materials, use of recycled materials, and use of eco-friendly materials, including organic materials.

However, regarding the follow-up question about the limitations, they focused on communicating to consumers these keywords, which represent the values regarding sustainable fashion. Answers such as "Environmental and social values are not the biggest reason for purchasing a product for consumers", "Difficulty in helping consumers understand the eco-friendly manufacturing process", and "Difficult to find keywords that fit their viewpoint" were reported. There were opinions that there are small changes in public opinion. However, concerning the growth of a sustainable fashion brand, it was found necessary to create an environment where the core values of sustainable fashion practiced by designers can be recognized/used more widely.

4.1.3. Awareness, Necessity, and Limitations of Sustainable Fashion Certification Indices

This study investigated designers' perceptions of the certification indices used in the fashion industry and the status of the brands that use them. There were the following types of international certifications recognized by designers.

Designers are most familiar with certifications at the raw material and material acquisition stage; eco-friendly textile certifications such as Global Organic Content Standard (GOTs); recycled raw material certifications such as Recycled Claim Standard (RCS) and Global Recycled Standard (GRS); animal protection certifications such as Responsible Wool Standard and PETA-approved vegan; and hazardous substances-related certifications such as the International Association for Research and Testing in the Field of Textile and Leather Ecology (OEKO-TEX). There is also the Better Cotton Initiative (BCI), a certification for training and supporting farmers in the production of sustainable cotton in the production phase, the Forest Stewardship Council (FSC), a certification for the protection of forests, and the World Fair Trade Organization (WFTO), a certification for fair trade, as well as ECO-CERT, which offers various certifications according to environmental and social standards such as organic, environmentally friendly textiles and fair trade. The Higg Index and the Blue Sign were mentioned for certification that comprehensively considered environmental and social impacts across the value chain.

However, most of the certifications are received by material producers at the material production stage of the value chain. Only two brands received brand-level certification. These were "OPEN PLAN" that received certification by GOTs and "G:RU" recognized by the WFTO. Although each designer suggested that sustainable fashion requires an integrated improvement across the value chain, these findings demonstrated that it is unrealistic for a small social venture brand to receive objective certification owing to the costs, time, and procedural hurdles. Therefore, in the journey of improving sustainability throughout the value chain of the fashion industry, it is expected that the core values presented through this study can play a meaningful milestone.

*4.2. Status and Views on the Core Values of Sustainable Fashion*

Most designers selected the representative value of their brand within the ten core values suggested by the researcher without much difficulty (Table 4). However, even though the standards of the values were presented to the respondents, some designers did not check whether they were a women's company. There were also a few cases where they checked a value that deviated from the presented standard. First, most designers agreed that the ten core values suggested by the researchers almost encompassed the core values that could be explained in sustainable fashion. This can be confirmed by the fact that most respondents answered "No" or "I cannot think of another keyword" to a question asking for other keywords. However, in the case of some designers, there was an opinion that gender is a more expanded concept than the keyword "women's company". Moreover, there was an opinion that donation/sharing is an additional concept that is not encompassed by the

value chain. It was suggested that the keyword "labor human rights" should be added. Contrary to the opinion of the designers presented in this study, as donation/sharing has a meaning of returning profits from an economic point of view, it will be preserved as a major core value. Labor human rights will not be added as a new value because it is a value that has already been directly or indirectly included in the detailed standards of local production or fair trade.

**Table 4.** Status of core value implementation of sustainable fashion brands in Korea.

| Brand | Main Category | Core Environmental Values | | | | | Social Core Values | | | | |
|---|---|---|---|---|---|---|---|---|---|---|---|
| | | Eco-Friendly | Recycled | Vegan | Upcycled | Zero Waste | Local Production | Fair Trade | Women-Owned | Gives Back | Animal Protection |
| G:RU | Women's wear | ○ | | | | | ○ | ○◎ | ∨ | | |
| ZERO DESIGN | Clothing, Misc. goods | | ○ | | ○ | ○◎ | ○ | | ∨ | | |
| CUE CLYP | Misc. goods | | ○ | | ○◎ | ○ | ∨ | | ∨ | | |
| OPEN PLAN | Women's wear | ○ | ○ | ○◎ | ○ | ○ | ○ | △ | ∨ | | ○ |
| NOT OURS | Clothing, Misc. goods | ○ | ○ | ○◎ | | | ○ | | ○ | | ○ |
| Harlie K. | Misc. goods | | | | ○◎ | | ○ | | ○ | ○ | |
| OVER Lab. | Misc. goods | ○ | ○ | | ○◎ | | ○ | | ○ | | |
| I WAS PLAS-TIC | Misc. goods | | ○◎ | | | ○ | | | ∨ | ∨ | |
| ICONPLE | Women's clothing | ○◎ | ○ | | | | ○ | | ○ | ○ | |
| Project 1907 | Misc. goods | | ∨◎ | | △ | ○ | ∨ | | | | |

○: The core values of the brands checked by designers. ◎: Representative core values of each brand that the researchers judged through the preliminary research and interview process. ∨: Values the designers did not check, but according to the detailed criteria presented by the researcher, core values that correspond to the brand. △: Values the designers checked, but core values that are not applicable according to the detailed criteria presented by the researcher.

A question was raised about whether it is appropriate to use the keyword "eco-friendly" and whether it should be used equally with the other core values. Here, "eco-friendly" refers to "environmentally friendly materials among materials derived from nature", such as any organic things, ethically produced wool, or eco-friendly regenerated fibers, indicating a clear difference from other core values. However, because it is not easy to name it in one word, in this study, the word "eco-friendly" was suggested; we are still looking for a more appropriate word. Vegetable processing using natural tannins such as tree bark was one of the detailed standards of vegan, and some respondents pointed out that it cannot be called vegan because it means using animal skins. Thus, the detailed standards were revised to reflect this opinion. There were opinions that keywords such as customizing, on-demand production, zero waste production, and custom-made should be added to the ten core values. Instead of adding the suggestions as core values, the study was added as a detailed standard for the zero-waste value in the future.

In addition, the cases where the respondents did not look closely at the standards presented or cases where the respondents had differing opinions on the standards presented are as follows. According to the detailed criteria presented, nine designer brands were women's companies, but five designers submitted their answers without checking the

criteria. The study did not ask further questions in this regard, but it was judged that the fact that they were women's companies did not give much meaning to the core value of sustainable fashion. It is thought that maintaining women's businesses as the core value of sustainable fashion is a meaningful approach as part of an effort to contribute to the 50:50 contribution of women's social roles to society. However, there is a higher proportion of females in the fashion industry than in other industries. Thus, it may be necessary to adjust the detailed criteria in the future. It is also possible to redefine gender by considering it an expanded concept.

Another example of an unclear answer was shown in the semi-structured survey. There were criteria for "recycled" and "upcycled", but despite being a "recycled" brand, some respondents submitted an "upcycled" brand. The terms "recycled" and "upcycled" are often used interchangeably, even among industry experts. However, as defined by the SUP (Seoul Upcycling Plaza) [43], recycled is defined as making a material reusable after returning a product to the raw material stage by applying physical/chemical transformations such as pulverization/crushing. Upcycled is defined as maintaining the traces of used material as much as possible and sublimating it into a new design product. This distinction will be maintained in this study as well. Besides that, the suggested two or three of the core values contain vague standards. Thus, there are core values that need to be clarified in their definitions. The areas that require additional research to build the detailed standards of core values in the future are as follows: the distinction between eco-friendly and vegan values; the renaming of eco-friendly values; the distinction between giving back and animal protection (so that the respondent can choose only one); and the redefinition of women-owned.

### 4.3. Proposal of Core Values and Detailed Standards for Sustainable Fashion

Based on the above in-depth interviews with domestic sustainable fashion experts, the core values and detailed standards of sustainable fashion were specified and presented as practical values and practices that can be communicated to consumers (Table 5). It demonstrates the specific details of the implemented core values as final standards (re-established detailed standards) that have undergone changes such as correction, removal, and confirmation through expert interviews.

### 4.4. The Case of the Establishment of Market Gredit: A Sustainable Fashion and Lifestyle Platform in Korea

Based on the derived core values and detailed standards Table 5 of sustainable fashion, this study launched an online commerce platform, Market Gredit, for sustainable fashion and lifestyle products. Market Gredit is a currently operating value consumption commerce platform (Figure 2) that carefully selects brands and products according to established detailed standards (Table 5) through the research process described above, which consists of five core environmental values and their corresponding representative icons. They are eco-friendliness, recyclability, being vegan, upcyclability, and production of zero-waste. It also focuses on five social values with a representative icon for each, in which the socially vulnerable participate, such as local production, fair trade, women-owned, giving back, and animal protection. The Market Credit website is currently only in Korean, but a Korean English version is being planned for global consumers. Figure 2 describes the definitions for each of the ten core values in Table 5 in Korean for South Korean consumers. In the brand and product selection stages, corporate research through various routes, face-to-face networking, and non-face-to-face interviews was conducted [44]. It is operated only to attract brands and products that coincide with at least one in each area of environmental and social core values. In other words, Market Gredit recruited brands that faithfully implemented at least some of these sustainable practice standards through the platform and promoted the brand and products so that consumers could understand them well.

**Table 5.** Core values and detailed standards of sustainable fashion.

| Core Values | | Existing Detailed Criteria | Re-Established Detailed Standards |
|---|---|---|---|
| Environ-mental values | **Eco-friendly** | - Natural fibers<br> • organic cotton, hemp, and ethically produced wool<br> • eco-friendly regenerated fibers such as Tencel<br>- (vegetable) Natural dye | - use of materials classified as eco-friendly among materials that are organic, natural, and nature-derived materials<br> • Natural fibers such as organic cotton, hemp, and wool<br> • Regenerated fibers such as Tencel<br>- (vegetable) natural dye<br>- However, it does not overlap with the vegan value. |
| | **Recycled** | - Recycled polyester or nylon<br>- Dyeing techniques that reduce water use, such as E –dye<br>- Recyclable after disposal | - Material returned to the raw material stage by applying physical/chemical transformation<br>- Materials that are easy to recycle after disposal |
| | **Vegan** | - Owner leadership to practice vegan life<br>- Use of non-animal eco-friendly fibers<br>- (For leather) vegetable processing<br>- (For cosmetics) Exclusion of animal testing | - Owner leadership to practice vegan life<br>- Non-animal eco-friendly material<br>- (For cosmetics) Exclusion of animal testing<br>- However, in the name of being a non-animal product, the use of plastic materials such as artificial leather is not permitted. |
| | **Upcycled** | - Upcycling design/production using waste materials<br>- (it is different from recycling in that it is designed/manufactured by preserving the original materials) | - Upcycling design/manufacturing using waste materials<br>- (different from recycling in that design/manufacturing is done using the raw waste materials) |
| | **Zero waste** | - Product: pattern method with less than 5% waste<br>- Packaging: plastic-free packaging | - Product: pattern method with less than 5% waste<br>- Packaging: plastic-free packaging |
| Social values | **Local production** | - Local production (through the socially vulnerable class or social economy organization) | - Local production (through the socially vulnerable class or social economy organization) |
| | **Fairtrade** | - Third World Fair Trade Products (through support for economic independence and sustainable development of producers in developing countries) | - Third World Fair Trade Products (through support for economic independence and sustainable development of producers in developing countries) |
| | **Women-owned** | - Female ownership, Companies with 50% or more women<br>- Companies employing women from disadvantaged groups | - Female ownership, Companies with 50% or more women<br>- Companies employing women from disadvantaged groups |
| | **Gives back** | - Regular sponsorship to organizations that attempt to solve environmental and social problems | - Regular sponsorship to organizations that attempt to solve environmental and social problems<br>- However, it does not overlap with the value of animal protection |
| | **Animal protection** | - Supports activities such as regular sponsorship of animal protection groups and leadership that practice vegan life | - Supports activities such as regular sponsorship of animal protection groups and leadership that practice vegan life |

As a result, we were able to attract 29 brands (16 for fashion and 13 for lifestyle) and 200 products, and we will continue to expand the pool. Figure 3 illustrates the status of brands in Market Gredit that continue to practice the core values of fashion as of May 2022. Figure 3 is a screen capture where South Korean consumers can check the list of brands and the representative core values of each brand with symbols. On each product detail page, the representative sustainable core values applied to that product are marked with icons and detailed descriptions that consumers can easily review. Figure 4 is a brand called Iconple that presents a transformable weather trench coat, and the brand practices the three core values such as "eco-friendly" with recycled nylon, "local production", and "gives back" as a member of "1% for the Planet". The ten core values are meaningful in that they not only allow for careful selection of products from valued manufacturers, but also provide criteria for selecting sustainable fashion that is easy to understand even by value-oriented consumers who have visited Market Gredit. The core value of Market Gredit serves as a standard for Market Gredit to carefully select sustainable brands and products and, at

the same time, as a tool to communicate with consumers. Although Market Gredit is a social venture, since it is a for-profit brand, it receives commissions from the brands sold on the platform. Market Gredit signs a contract with a store brand on an annual basis and, at the end of each year, decides to renew the contract based on an evaluation of whether they still meet the store criteria according to the key implementation status of the core sustainable values. In addition, Market Gredit mainly communicates its philosophy for platform operation through social media channels and newsletters and strives to meet consumers who understand it. These values are expected to contribute to spreading a culture of right-value consumption.

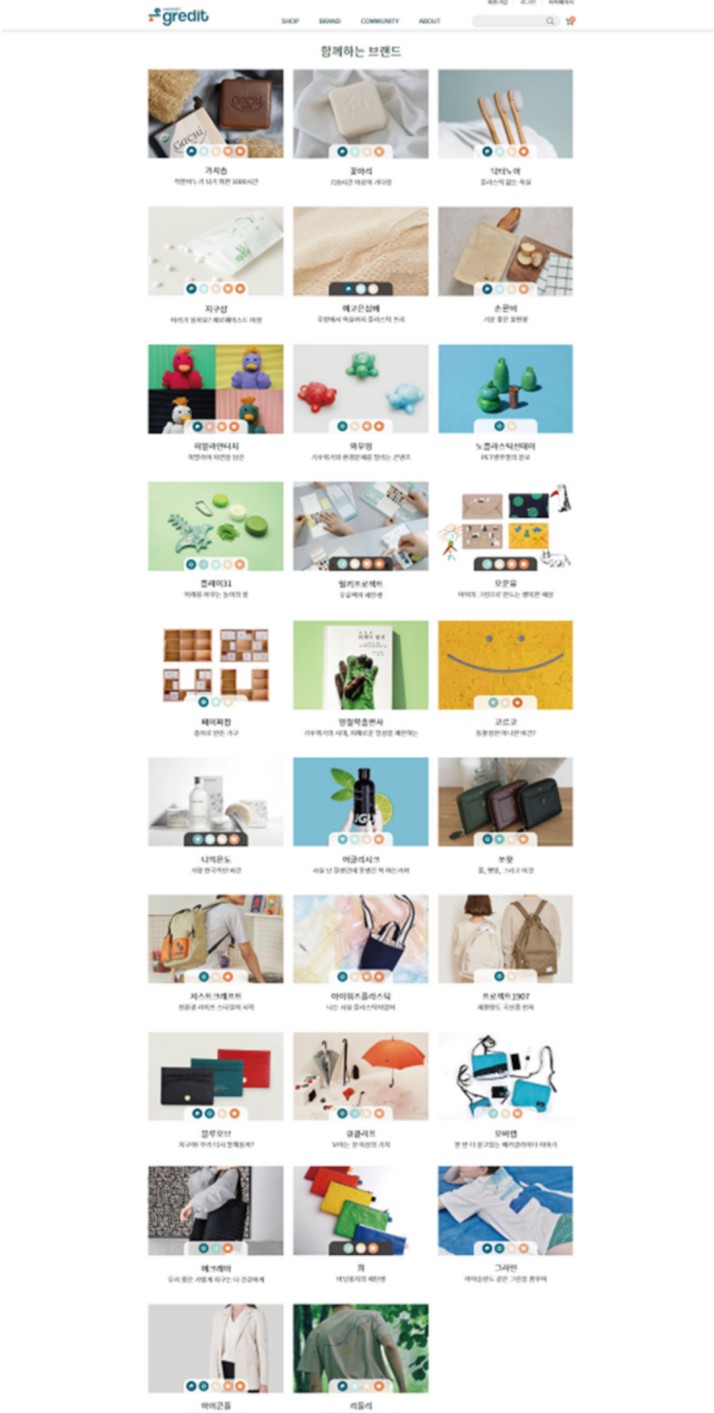

**Figure 3.** Brands that are part of Market Gredit. Source: https://gredit.io/custom/common/brand.html?cate_no=190 (accessed on 10 July 2022).

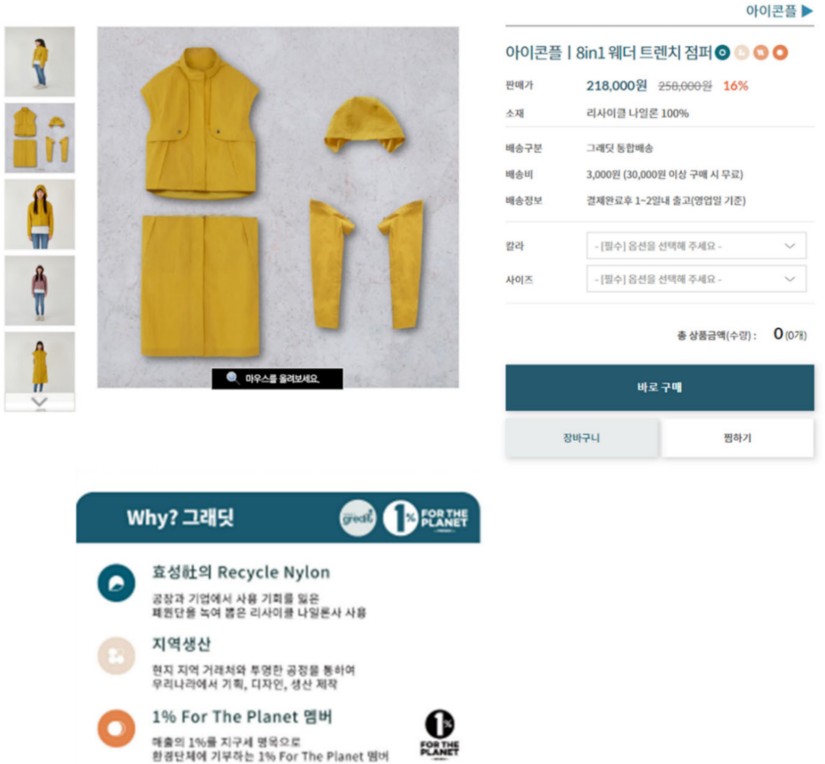

**Figure 4.** Example of core values applied to a product in Market Gredit. Source: https://gredit.io/product/%EC%95%84%EC%9D%B4%EC%BD%98%ED%94%8C%E3%85%A38in1-%EC%9B%A8%EB%8D%94-%ED%8A%B8%EB%A0%8C%EC%B9%98-%EC%A0%90%ED%8D%BC/188/category/212/display/1/ (accessed on 2 May 2022).

## 5. Conclusions

The results of the study contribute to improving the environmental and social impact across the entire process of the value chain of the fashion industry. By proposing the core values and detailed standards of sustainable fashion suitable for communication with consumers, this study provides academic, practical, and educational implications. First, academically, by conducting qualitative research based on in-depth interviews with domestic sustainable fashion experts, this study verified the validity of the core values and practices of sustainable fashion. The core values proposed in this study were applied only to creating Market Gredit. However, through follow-up research, additional expert verification for each core value will be carried out, and quantitative standards for detailed standards will be secured so that they can be used in related industries with more objective public confidence in the future.

Second, this study has significance as it proposes realistic core values that can improve the environmental and social practices of sustainable fashion designers and producers and ultimately communicate with consumers in the fashion industry. On this basis, it suggests establishing a platform where domestic brands with high-level sustainability practices operate. This is expected to expand the positive cycle of sustainable domestic fashion by directly or indirectly expanding the number of brands that participate in platforms and raising the level of sustainability of participating brands. Market Gredit operates social media channels that introduce brands and products with the ten core values. Through this, it aims to revitalize the market by helping consumers more easily understand sustainable fashion, increase access to sustainable fashion brands, and spread a value consumption culture. In particular, as Generation Z is highly conscious of eco-friendly value consumption and wants accurate product information, introducing brands and products through sustainable core values is expected to increase consumer understanding and motivate purchases.

Third, in the educational setting, the core value of sustainable fashion is expected to help students understand the sustainable fashion industry from a practical perspective. It can be used as essential guidelines when developing designs and brands. In addition, it will serve as a standard for understanding the sustainable practices of various brands that advocate sustainable fashion. It will serve as meaningful basic data for students to explore careers and entrepreneurship in related industries and to develop their competencies.

This study has several limitations. First, due to the complex structure of the fashion supply chain, some of the ten core values proposed in this study overlap or have unclear distinctions. For example, there is an overlap in detailed standards between eco-friendly plant material and vegan material, as well as between giving back and sharing/donation activities of animal protection associations. Second, one of the core values, i.e., eco-friendliness, should be supplemented with a term that can convey a clearer meaning. Moreover, the keyword, i.e., "women-owned", needs to be supplemented by presenting standards from a gender point of view. It seems necessary to have deeper studies that will create more sophisticated and detailed criteria to communicate with the public. Third, the ten core values proposed in the study can be used as more meaningful indicators when quantitative standards such as credible authentication or the Higg index are presented together. Although there is currently no correlation between the actual number of core values and the degree of quantitative improvement in environmental friendliness, a large number of core values is often misunderstood as having high environmental and social practices. Finally, the purpose of this study is to derive sustainable keywords through the analysis of the status of producers and communicate to meet the consumer's eye level, but there is a limitation as this paper focuses on the former. These limitations will be supplemented through consumer research in follow-up studies, and consumers' understanding and empathy, and the effectiveness of the core sustainability values and Market Gredit platform will be confirmed. Consumer research and analysis must continue to verify how the core values of sustainable fashion provided by Market Gredit and various related content are helping consumers understand sustainable fashion.

This study aims to prepare standards and implementation details (action items) that can communicate with consumers from a producer's point of view regarding the core values of sustainable fashion and establish a basis for verifying and actively implementing the sustainable values through the case of South Korea's social venture platform. Moreover, it is expected to serve as an opportunity to discover various sustainable brands in South Korea through the fashion lifestyle platform.

**Author Contributions:** All processes of this research have been conducted together by S.S. and Y.K. All authors have read and agreed to the published version of the manuscript.

**Funding:** This work was supported by the 2021 Gachon University research fund (GCU-202103540001).

**Institutional Review Board Statement:** The study was approved by the Bioethics Committee of Gachon University (1044396-202109-HR-187-01).

**Informed Consent Statement:** Informed consent was obtained from all subjects involved in the study.

**Data Availability Statement:** Not applicable.

**Conflicts of Interest:** The authors declare no conflict of interest.

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
