# Peer review of "The Core Value of Sustainable Fashion: A Case Study on “Market Gredit”"

_sustainability, doi:10.3390/su142114423_

Round 1

Reviewer 1 Report

The paper describes the study on deriving core value(s) of sustainable fashion that would penetrate through the entire process: from sourcing raw materials and manufacturing to design, production, sales and distribution.   

While the topic per se is very relevant to the scope of the journal, I have some questions (doubts) regarding research design and particularly the choice of methods. I would not solely rely on interviews as a convincing method of collecting and understanding opinions qualitatively. As Kohtala and Hyysalo* brilliantly shown, what makers say about sustainability is not equal to what they actually do. There are a lot of “greenwashing” cliches, usually embedded in people’s speech with little or no reflection at all. 

Also, what was the language of interviews? And what was the language of introducing ten keywords? If English, how did you make sure that you and your informants had the same understanding? How could you guarantee no ‘loss in translation’?

* Kohtala, C., & Hyysalo, S. (2015). Anticipated environmental sustainability of personal fabrication. Journal of Cleaner Production, 99, 333-344.

Next, I recommend to redesign tables and diagrams – to make them more compact and clearer (adjust font sizes, interline spacing, etc.) – as for now they are hard to read and navigate through.

Some specific points and comments:

Line 495-496: Did I get it right: you interviewed design employees, not brand owners? It sounds weird that some of them did not even know who owns the brand, and itrevokes even more questions about your choice of respondents – whether their other answers are adequate to what is actually going on in the companies.

Line 498-502: There is a false assumption in this paragraph:

“First, most designers agreed that the ten core values suggested by the researchers almost encompassed the core values that could be explained in a sustainable fashion. This can be confirmed by the fact that most respondents answered “No” or “I cannot think of another keyword” to a question asking for other keywords.”

You conducted a complicated literature search, spent a lot of time analyzing it to come up finally with particular concepts. And now you are asking people who rarely work with words and sophisticated concepts to suggest yet another concept, and got surprised when they did not come up with anything new. Isn’t it strange?

Table 5: what’s the point to show criteria that were not reconsidered? Why not to highlight changes in values and concepts explicitly?

Regarding the Market Gredit: What are the conditions for brands participated in the platform? Should they pay? Should they go through a kind of regular evaluation to confirm they still fit?

In the description of the platform, there is no answer to the question of how to promote this platform to potential consumers (domestic and abroad). I did not clearly understand what the difference to other existing forms and interaction arenas was in the eyes of potential customers.

Line 608-610: I did not see any evidence of the connection, clear understanding and sharing of values that allegedly exist between fashion designers and domestic consumers. Overall, the study describes no connection with consumers whatsoever.

You can find my other comments and suggestions in the file attached.

Finally, I do hope the authors could rework their paper as it has some relevance and promise and can potentially become a valuable contribution. In terms of rework, I would also recommend thinking about what specific is about South Korea in terms of understanding and promoting sustainable fashion on both domestic and global markets.

Author Response

Thank you so much for reviewing my research with your interest and constructive feedback. We attached the revised version of the manuscript and the response letter to your comments.

Reviewer 2 Report

Please see my comments attached.

Author Response

(The authors gave the same response as above.)

Reviewer 3 Report

Figures use for the Market Gredit interface in the journal is in Korean language. Since the website can be viewed in English version, it is good to suggest that figures used (Figure 2, 3, and 4 at page 17 until 19) to use English language to enhance readers engagement while read this paper. 

Author Response

(The authors gave the same response as above.)

Reviewer 4 Report

The paper presents an interesting and relevant research that is within the scope of the Sustainability journal. 

The manuscript reports a qualitative case study that explores ten core environmental and social values of sustainable fashion. It encompasses a comprehensive literature review, in-depth interviews that analyze how value producers pursue sustainable fashion, as well as the application of the studied values to an e-commence platform for sustainable fashion brands.

The outcomes are relevant and the limitation of the study are well discussed.

I have some suggestion for the authors’ appreciation:

p.2 line 64-67: References should be added that support this phrase.

Table 5: Fair trade value could give more descriptive information towards consumer communication.

4.4 Section: It was not clear for me who defines the core values for each product, is it the brand, the Market Gredit or both?

Overall recommendation: minor revision

Author Response

(The authors gave the same response as above.)

Round 2

Reviewer 1 Report

I would like to thank the authors for the work done: the manuscript has been improved significantly. I wish them all the best with their further research and practice in the field of establishing and promoting sustainability.